

# Effects of weight-bearing dance aerobics on lower limb muscle morphology, strength and functional fitness in older women

Xiaoying Peng[1,*], Tang Zhou[2,*], Hua Wu[3], Yiyan Li[4], Jiajia Liu[1], Huan Huang[1], Changshuang He[1], Shaoyu Guo[1,5], Muyang Huan[1], Lei Shi[1], Peijie Chen[1] and Minghui Quan[1,6]

[1] School of Exercise and Health, Shanghai University of Sport, Shanghai, China
[2] Pinghu Normal University, Jiaxing University, Jiaxing, China
[3] Rehabilitation Medicine Center, The Second Affiliated Hospital of Jiaxing University, Jiaxing, China
[4] Shenzhen Longhua School Affiliated to East China Normal University, Shenzhen, China
[5] High School Affiliated to Fudan University, Shanghai, China
[6] Shanghai Frontiers Science Research Base of Exercise and Metabolic Health, Shanghai, China
* These authors contributed equally to this work.

## ABSTRACT

**Objective:** To investigate the effects of 12-week weight-bearing dance aerobics (WBDA) on muscle morphology, strength and functional fitness in older women.
**Methods:** This controlled study recruited 37 female participants (66.31y ± 3.83) and divided them into intervention and control groups according to willingness. The intervention group received 90-min WBDA thrice a week for 12 weeks, while the control group maintained normal activities. The groups were then compared by measuring muscle thickness, fiber length and pennation angle by ultrasound, muscle strength using an isokinetic multi-joint module and functional fitness, such as 2-min step test, 30-s chair stand, chair sit-and-reach, TUG and single-legged closed-eyed standing test. The morphology, strength, and functional fitness were compared using ANCOVA or Mann-Whitney U test to study the effects of 12 weeks WBDA.
**Results:** Among all recruited participants, 33 completed all tests. After 12 weeks, the thickness of the vastus intermedius ($F = 17.85$, $P < 0.01$) and quadriceps ($F = 15.62$, $P < 0.01$) was significantly increased in the intervention group compared to the control group, along with a significant increase in the torque/weight of the knee flexor muscles ($F = 4.47$, $P = 0.04$). Similarly, the intervention group revealed a significant improvement in the single-legged closed-eyed standing test ($z = -2.16$, $P = 0.03$) compared to the control group.
**Conclusion:** The study concluded that compared to the non-exercising control group, 12-week WBDA was shown to thicken vastus intermedius, increase muscle strength, and improve physical function in older women. In addition, this study provides a reference exercise program for older women.

Corresponding authors
Peijie Chen, chenpeijie@sus.edu.cn
Minghui Quan,
quanminghui@163.com

## INTRODUCTION

A healthy physical function determines good life quality during later life (*Studenski et al., 2003*), promotes an extended life expectancy among the older (*Katz et al., 1983*), and helps reduce mortality (*Studenski et al., 2011*). Muscle mass and strength are pivotal for preserving physical function in the older (*Gariballa & Alessa, 2013*; *Landi et al., 2012*). It has been reported that muscle strength typically decreases by approximately 30% with aging (*Aoyagi & Shephard, 1992*). In contrast, the muscle cross-sectional area experiences a reduction of approximately 40% within 50 years, spanning from 20 to 70 years of age (*Rogers & Evans, 1993*). Furthermore, the transition to menopause triggers hormonal changes in women, leading to a decline in muscle mass and grip strength (*Messier et al., 2011*). Women have a longer life expectancy than men, so their susceptibility to muscle loss may increase (*Javed et al., 2019*). Therefore, it is crucial to pay attention to the issues of muscle loss and muscle strength decline among older women.

Physical activity (PA) is a potent protective strategy for delaying muscle loss and muscle strength decline in the older (*Cruz-Jentoft et al., 2014*). It is recommended that older adults engage in different physical activities, including resistance exercises and balance training (*Cruz-Jentoft et al., 2019*; *Piercy et al., 2018*). A prospective study reported that weight-bearing exercise, a resistance exercise using the body weight as a load or introducing external weights through vests or pockets, enhanced muscle strength in premenopausal women (*Rockwell et al., 1990*). In existing studies, the forms of weight-bearing exercise incorporated different training components. The studies lasted six weeks to one year, and the attendance was relatively high, from 68% to 97%, despite differences in training approaches (*Baggen et al., 2019*; *Kim et al., 2018*; *Mair, De Vito & Boreham, 2019*; *Marques et al., 2011*; *Winters-Stone et al., 2014*). However, this form of exercise is not as popular as expected in China.

Square dancing in China is popular among older women (*Chen et al., 2022*). Dance aerobics combines rhythmic movement with controlled breathing, requiring the engagement of muscular strength, endurance, balance coordination, cardiopulmonary endurance and cognitive involvement. It was reported that dancing can significantly improve body composition and physical function (*Fong Yan et al., 2018*), and reduce the risk of hypertension and coronary heart disease (*Jia et al., 2018*). Square dancing also has a significant positive effect on psychology (*Bai et al., 2022*).

Since weight-bearing exercise amplifies training intensity and augments skeletal muscle stimulation, dance aerobics can heighten interest and engagement among older women. Therefore, amalgamating these dual exercise modes can address the limitations of singular training approaches and enhance muscle mass, strength, and physical fitness.

The present study aimed to explore the impact of the combination of dance aerobics and weight-bearing exercises on lower limb muscle morphology, strength, and functional fitness in older women. It is hypothesized that compared to the non-exercising control group, 12-week weight-bearing dance aerobics (WBDA) will yield augmented lower limb muscle dimensions, increase muscle strength, and improve functional fitness among older women.

## METHODS

### Participants

Thirty-seven older women were recruited through advertising posters in the community and online platforms in the Yangpu and Hongkou districts of Shanghai. During the first appointment, questionnaires were completed by the participants, which included the physical activity readiness questionnaire (PAR-Q) for pre-screening, the adapted version of the international physical activity questionnaire-short form (IPAQ-SF) to assess physical activity, including sedentary behavior (SB), light physical activity (LPA), moderate and vigorous physical activity (MVPA), total physical activity (TPA), and included the intake of calcium supplements and vitamin D. We enrolled all participants who met the following inclusion criteria: 1) women aged 60–75 years old; 2) a relatively low amount of physical activity (in the past 3 months, moderate-to-vigorous exercise less than three times weekly with a session lasting less than 30 min); 3) those who participated voluntarily and signed informed consent. The exclusion criteria were as follows: 1) individuals with any potential health risks associated with exercise (confirmed by PAR-Q) or advised against participating by medical professionals; 2) those with severe diseases, such as severe heart disease, diabetic neuropathy respiratory disease, orthopedic conditions, post-surgical recovery, unstable blood pressure, or other conditions which prevented them from completing the exercise and physical fitness test.

All participants signed informed consent. The study was approved by the Ethics Committee of Shanghai University of Sport (ethic committee code: 102772020RT096), and registered by the China Clinical Trials Center (ChiCTR2100047187).

### Intervention

Thirty-seven participants were divided into two groups based on their preference, with 20 participants in the intervention group, and 17 in the control group. Participants in the intervention group performed 12 weeks WBDA three times a week, each session 90 min, including 10 min of warm-up, 60 min of exercise, 10-min stretching, and two 5-min rest periods. The music accompanied the exercise routine set at approximately 130 beats per minute. The subjects were engaged in dance aerobics, including but not limited to march, walk, easy walk, mambo, V-step, grapevine, step touch, leg curl, and knee lift-up.

All participants wore a weighted vest, net weight 750 g, and the vest pockets were loaded with 150 g lead plates prescribed as a percentage of their body weight. Participants were freely acquainted with dance aerobics (DA) in the first week without load. Subsequently, the load was systematically increased, as listed in Table 1.

To adjust the intensity of exercise promptly and to prevent potential accidents, subjects wore a heart rate monitor (Polar OH1) to monitor the real-time heart rate. Exercise intensity was formulated as 60–75% of the maximum heart rate (*Pescatello, 1999*), where age-predicted maximum heart rate = 220–age.

**Table 1 Weight adjustment for weighted vests.**

| Time | Week 1 | Weeks 2–5 | Weeks 6–12 |
|---|---|---|---|
| Vest load | 0 | 2% body weight | 5% body weight |

## Measurements

### Measurement of muscle morphology

Muscle thickness, fiber length, pennation angle of the rectus femoris (RF), vastus intermedius (VI), soleus, medial gastrocnemius (GM), and lateral gastrocnemius (GL) in the dominant leg were measured using a portable ultrasound device (LOGIQ e, US, L4-12t).

The RF and VI were measured at the midpoint between the femur's greater trochanter and lateral condyle, with the subject in a supine and relaxed position. Soleus, GL, and GM muscles were measured at the proximal 1/3 between the medial femoral condyle and the medial ankle, with the body in the prone position. Each muscle was measured three times, and the results were averaged.

### Measurement of muscle strength

Peak torque and endurance of the dominant leg's knee flexor and extensor muscles were measured using an isokinetic multi-joint module (CON-TREX-MJ, PHYSIOMED, Bayern, Germany). The isometric strength testing protocol was as follows: 1) when testing peak torque, the participants were tested over five repetitions at 60°/s hip or knee flexion and extension; 2) when testing endurance, the participants were tested over 20 repetitions at 180°/s knee flexion and extension.

### Measurement of functional fitness

#### 2-min step test

The two-minute step test reflected aerobic endurance. Participants were instructed to raise their knees to a height equivalent to the midpoint between the anterior superior iliac spine and the patella. In the event of failure to raise knees to a specified height or to do so on only one side, they were not counted. Participants were allowed to pause, rest, and restart until completion of the 2-min duration.

#### 30-s chair stand (30 s CST)

Lower body strength was evaluated by 30 s CST. Participants were positioned in the center of a 43 cm high chair, with their feet resting flat on the ground and their arms crossed in front of their chests. Upon the command "start," participants were required to stand upright and return to a complete sitting position. The number of times the participants completed the entire stand-sit cycle within a 30-s interval was recorded.

#### Chair sit-and-reach

Lower limb flexibility was evaluated using chair sit-and-reach (CSR). The seated participants were made to extend their one leg and bend forward slowly, sliding their hands down the extended leg to touch (or past) the toes. The score was defined as the

distance between the tip of the middle finger and the toe. The score of 0 was assigned if the participant succeeded in touching the toe, while if the middle fingers could not, the score was marked as a negative number (−). Similarly, if the middle finger extended beyond the tip of the foot, the distance score was recorded as a positive number (+). All measurements were taken twice, with the maximum distance recorded as the final score.

*Time up and go*
Time up and go (TUG) reflects dynamic balance and walking ability. Participants were required to sit in a 45 cm high chair, walk a 3 m course as fast as possible, walk back to the chair, and sit again. The time taken was measured in seconds. Each participant was tested twice, and the average was recorded.

*Single-legged closed-eyed standing test*
Single-legged closed-eyed standing test (SCST) reflects the subject's static balance. The participants were made to stand with their eyes closed, and when hearing the command "start," they had to lift one foot off the ground. The test was terminated when the supporting foot moved, or the raised foot touched the ground. The results were rounded to two decimal places and measured in seconds.

### Compliance and attendance rates
Attendance was recorded during thirty-six training sessions. Compliance was determined by comparing the number of people in the statistical analysis to the total number of people recruited. The attendance rate was defined as the number of exercise sessions reported divided by the number of maximum exercise sessions possible, and the calculation formula is as follows *Klentrou et al. (2007)*:

$$\text{Attendance rate} = \frac{\sum \textit{Number of sessions actually attended each person}}{36 * 18}. \tag{1}$$

## Statistical analysis
Independent samples t-test was performed to determine the differences at baseline. The paired-sample t-test was used to determine the differences between the participants' pre- and post-intervention muscle morphology, strength and functional fitness. Effect size, Cohen's d, was calculated and categorized as small (0.2 ≤ Cohen's d < 0.5), medium (0.5 ≤ Cohen's d < 0.8), or large (Cohen's d ≥ 0.8) effect sizes (*Cohen, 1988*).

With pre-test data as a covariate, ANCOVA was employed to compare inter-group differences. Effect size, partial $\eta^2$, defines small, medium, and large effect sizes as 0.01, 0.06, and 0.14 (*Cohen, 1992*). If the date violated the normal distribution (tested by the Shapiro-Wilk test), or had an interaction effect between the group and the baseline, the Mann-Whitney U test was used to analyze the effects of 12 weeks WBDA. The effect size (r) was calculated by dividing the Z-score by the square root of N, indicating a large effect size using Cohen's d criteria of 0.10, 0.30, and 0.50 to define small, medium, and large effect sizes, respectively (*Rosenthal, 1994*).

All statistical analyses were conducted using SPSS (version 26.0), with significant differences at $P < 0.05$ and borderline significance at $0.05 < P < 0.1$.

# RESULT

## Basic information

Among 37 recruited participants, two participants from each intervention and control group withdrew due to poor transportation and inability to train consistently. Thirty-three participants were included in the statistical analysis, with 18 in the intervention and 15 in the control group. The experimental process is shown in Fig. 1. Apart from age, no significant differences were observed between the two groups regarding height, weight, BMI, duration of menopause, and age at menopause. There were also no significant group differences in ST, TPA, LPA, or MVPA (Table S1). An overall compliance rate of 89.19% was observed, with the intervention group having a 90% compliance rate. The attendance rate of the intervention group was 95.99%. The average heart rate in the intervention group was 112.5 beats per minute, representing approximately 72% of the maximum heart rate (Table 2).

## The effects on muscle morphology

The morphological results of the knee extensor and flexor muscles are shown in Table 3. Compared to baseline values, after 12 weeks of intervention, the thickness of VI and quadriceps in the intervention group increased by $0.27 \pm 0.44$ cm ($P = 0.02$) and $0.40 \pm 0.61$ cm ($P = 0.01$), respectively. The RF had a trend of thickening (Cohen's d = 0.41, $P = 0.095$), with a reducing trend in RF pennation angle (Cohen's d = 0.42, $P = 0.09$). In contrast, the control group displayed a potential decline in the thickness of VI at the end of the 12-week duration (Cohen's d = 0.48, $P = 0.09$).

Moreover, compared to the control group, the VI and quadriceps thickness in the intervention group increased ($F = 17.85$, $P < 0.01$, partial $\eta^2 = 0.37$ and $F = 15.62$, $P < 0.01$, partial $\eta^2 = 0.34$), with a marginally significant increase in the thickness of soleus ($F = 3.59$, $P = 0.07$, partial $\eta^2 = 0.11$).

However, no significant difference was observed between the groups regarding the thickness of RF, soleus, GL and GM, the length of all muscle fibers and the pennation angle.

## The effects on muscle strength

The results of isokinetic muscle strength of the knee extensor and flexor muscles are shown in Table 4. The results indicated that after 12 weeks of intervention, the peak torque of knee flexors in the intervention group increased by $5.54 \pm 4.81$ Nm/kg compared with the pre-test, accompanied by a noteworthy increase in the peak torque/body weight ratio of $0.06 \pm 0.10$ Nm/kg. Furthermore, the knee peak torque flexors/extensors significantly increased by $7.21 \pm 10.43\%$. Conversely, no significant changes were observed in the control group.

Upon accounting for baseline differences ($P = 0.02$) through ANCOVA, a significant increase was observed in the intervention group in terms of flexors peak torque/body weight ratio after the 12-week intervention, in comparison to the control group ($F = 4.47$, $P = 0.04$, partial $\eta^2 = 0.13$).

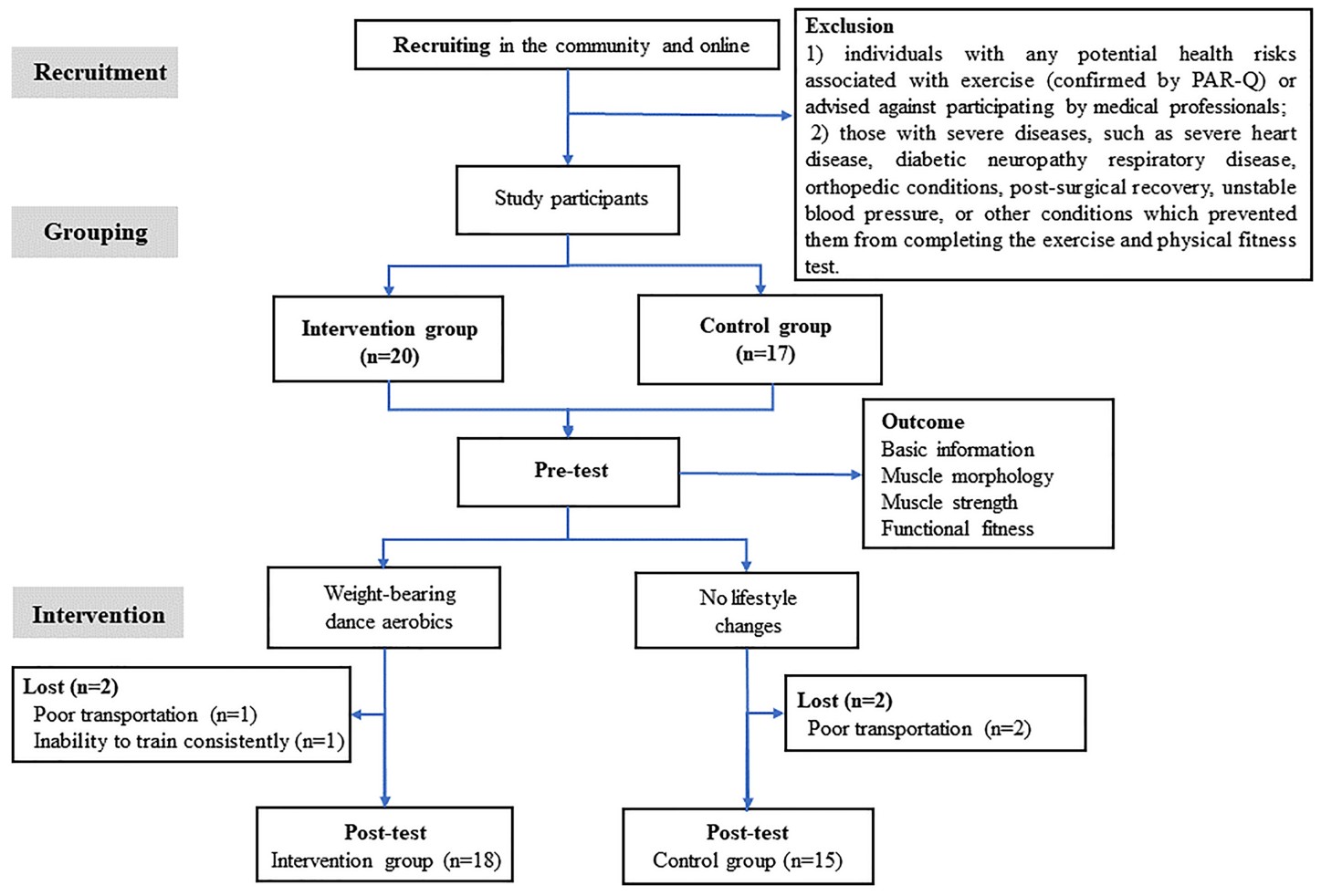

**Figure 1 Experimental process.**

| | Intervention ($n = 18$) (Mean ± SD) | Control ($n = 15$) (Mean ± SD) | $t$ | $P$ |
|---|---|---|---|---|
| Age (year) | 65.06 ± 3.64 | 67.67 ± 3.27 | −2.15 | 0.04 |
| Height (cm) | 157.46 ± 4.75 | 158.33 ± 4.42 | −0.54 | 0.60 |
| Weight (kg) | 58.17 ± 7.00 | 57.41 ± 5.60 | 0.34 | 0.74 |
| BMI (kg/m$^2$) | 23.45 ± 2.61 | 22.94 ± 2.50 | 0.57 | 0.57 |
| Menopausal age (year) | 50.72 ± 3.48 | 51.13 ± 2.56 | −0.38 | 0.71 |
| Compliance[*] | 90% | 88.24% | | |
| Attendance rates[#] | 95.99% | —— | | |
| Average heart rate (beat/minute) | 113.13 ± 3.43 | —— | | |

**Table 2 Basic information of the participant.**

**Note:**
BMI: Body Mass Index, BMI, height/weight$^2$.
[*] Attendance rates = 622/(36*18) = 622/648 = 95.99%.
[#] The attendance rate was defined as the number of exercise sessions reported divided by the number of maximum exercise sessions possible, namely attendance rates = 622/(36*18) = 622/648 = 95.99%.

**Table 3 The result of muscle strength.**

| | Intervention (*n* = 18) (Mean ± SD) | | Control (*n* = 15) (Mean ± SD) | | Between-group | | |
|---|---|---|---|---|---|---|---|
| | Pre | Post | Pre | Post | *F/z* | *P* | *Partial η2/r* |
| Muscle thickness (cm) | | | | | | | |
| RF | 1.40 ± 0.24 | 1.53 ± 0.25[b] | 1.29 ± 0.32 | 1.41 ± 0.28 | 0.81 | 0.38 | 0.03 |
| VI | 1.45 ± 0.39 | 1.71 ± 0.40[a] | 1.42 ± 0.37 | 1.23 ± 0.23[b] | 17.85 | <0.01 | 0.37 |
| Quadriceps[#] | 2.84 ± 0.55 | 3.24 ± 0.44[a] | 2.71 ± 0.65 | 2.64 ± 0.40 | 15.62 | <0.01 | 0.34 |
| Soleus | 2.08 ± 0.91 | 2.37 ± 0.82 | 1.68 ± 0.45 | 1.84 ± 0.52 | 3.59 | 0.07 | 0.11 |
| GM | 1.54 ± 0.26 | 1.65 ± 0.39 | 1.58 ± 0.23 | 1.58 ± 0.23 | 0.87 | 0.36 | 0.03 |
| GL | 1.20 ± 0.27 | 1.19 ± 0.16 | 1.22 ± 0.39 | 1.14 ± 0.23 | 0.72 | 0.40 | 0.02 |
| Length of fiber (cm) | | | | | | | |
| GM | 3.48 ± 0.50 | 3.61 ± 0.44 | 3.59 ± 0.52 | 3.74 ± 0.61 | 0.31 | 0.58 | 0.01 |
| GL | 3.77 ± 1.02 | 3.39 ± 0.61 | 3.39 ± 0.58 | 3.47 ± 0.50 | 1.15 | 0.29 | 0.04 |
| Pennation angle (°) | | | | | | | |
| RF | 10.66 ± 2.38 | 9.26 ± 2.51[b] | 9.53 ± 2.46 | 8.64 ± 2.28 | 0.20 | 0.66 | 0.01 |
| GM | 29.27 ± 3.32 | 29.02 ± 5.53 | 27.58 ± 4.58 | 28.59 ± 5.45 | 0.26 | 0.61 | 0.01 |
| GL[&] | 21.89 ± 4.49 | 24.20 ± 4.07 | 22.69 ± 3.74 | 22.54 ± 4.85 | −1.49 | 0.15 | 0.26 |

Notes:

RF, rectus femoris, VI, vastus intermedius, GM, medial gastrocnemius, GL, lateral gastrocnemius.

[#] The sum of the thickness of RF and VI.

[&] In cases where data did not conform to a normal distribution, and there was an interaction between the treatment factors and the pre-test data, Mann-Whitney U test was used to analyze the differences between the two groups pre and post-test change values. The effect size (r) was calculated by dividing the Z-score by the square root of N.

[a] post *vs.* pre, *P* < 0.05.

[b] post *vs.* pre, 0.05 < *P* < 0.1.

**The effects on functional fitness**

The results of functional fitness tests are shown in Table 4. The results showed that the SCST in the intervention group increased by 3.91 ± 5.35 s (*P* < 0.01) after 12 weeks of intervention, which was significantly higher than that in the control group (*z* = −2.16, *P* = 0.04).

Although the 30 s CST in the intervention group tended to increase (Cohen's d = 0.46, *P* = 0.07), a similar significant increase was observed in the control group (*P* < 0.01); thus, there was no significant difference between the two groups.

# DISCUSSION

The results of the 12-week exercise intervention indicated that WBDA could increase the thickness of the lower limb muscles and the strength of knee flexion muscles, and improve the static balance ability.

## Muscle morphology

### Muscle thickness

After 12 weeks WBDA, the thickness of VI increased in the intervention group, whereas a decreasing tendency was observed in the control group, which indicates that WBDA has a certain effect on increasing VI mass, potentially delaying muscle atrophy (*Rodriguez-Lopez et al., 2022*). It may be due to the significant involvement of the quadriceps in DA, such as

**Table 4 The result of muscle strength and functional fitness.**

| | Intervention (n = 18) (Mean ± SD) | | Control (n = 15) (Mean ± SD) | | Between-group | | |
|---|---|---|---|---|---|---|---|
| | Pre | Post | Pre | Post | F/z | P | Partial η2/r |
| **Peak torque (Nm)** | | | | | | | |
| Knee extensors | 62.34 ± 11.71 | 64.12 ± 9.43 | 56.28 ± 9.33 | 55.65 ± 11.80 | 2.81 | 0.10 | 0.09 |
| Knee flexors[&] | 35.35 ± 8.65 | 40.45 ± 7.66[a] | 27.17 ± 8.57[*] | 29.02 ± 8.23 | −1.92 | 0.06 | 0.33 |
| H: Q | 56.99 ± 10.8 | 64.04 ± 11.27[a] | 48.43 ± 14.25 | 55.76 ± 17.81 | 0.98 | 0.33 | 0.03 |
| **Peak torque/weight (Nm/kg)** | | | | | | | |
| Knee extensors | 1.10 ± 0.22 | 1.07 ± 0.19 | 1.00 ± 0.19 | 0.96 ± 0.22 | 0.97 | 0.33 | 0.03 |
| Knee flexors | 0.62 ± 0.16 | 0.68 ± 0.14[a] | 0.49 ± 0.16[*] | 0.48 ± 0.20 | 4.47 | 0.04 | 0.13 |
| **Endurance[#]** | | | | | | | |
| Knee extensors | 0.72 ± 0.10 | 0.74 ± 0.11 | 0.75 ± 0.09 | 0.82 ± 0.16 | 1.15 | 0.29 | 0.04 |
| Knee flexors | 0.85 ± 0.10 | 0.85 ± 0.11 | 0.84 ± 0.09 | 0.86 ± 0.19 | 0.01 | 0.94 | 0.01 |
| **2-min step (times)** | 126.34 ± 19.31 | 130.89 ± 13.49 | 99.71 ± 16.34[*] | 111.60 ± 15.04[a] | 1.68 | 0.21 | 0.05 |
| **30 s CST (times)** | 26.5 ± 3.84 | 28.79 ± 3.93[b] | 21.29 ± 6.12[*] | 24.53 ± 6.17[a] | 0.53 | 0.47 | 0.02 |
| **CSR(cm)** | 3.62 ± 8.24 | 8.37 ± 13.08 | 13.65 ± 13.51 | 11.40 ± 12.73 | 1.87 | 0.18 | 0.06 |
| **TUG (s)** | 5.59 ± 0.43 | 5.45 ± 0.62 | 5.81 ± 0.47 | 5.82 ± 0.63 | 1.46 | 0.24 | 0.05 |
| **STCT (s)[&]** | 5.54 ± 4.38 | 9.45 ± 6.84[a] | 5.60 ± 3.51 | 7.01 ± 6.14 | −2.16 | 0.03 | 0.39 |

Notes:
H, Q denotes flexors peak torque/extensors peak torque; 30 s CST, 30 s chair stand; CSR, chair sit-and-reach; TUG, timed up and go; STCT, single-legged closed-eyed standing test.
[#] Endurance refers to the ratio of the peak torque of the last six stretches (flexion) to the first six stretches (flexion) in the 20 flexions and extension of the endurance test.
[&] In cases where data did not conform to a normal distribution, and there was an interaction between the treatment factors and the pre-test data, Mann-Whitney U test was used to analyze the differences between the two groups' pre and post-test change values. The effect size (r) was calculated by dividing the Z-score by the square root of N.
[*] Intervention vs. control, $P < 0.05$.
[a] post vs. pre, $P < 0.05$.
[b] post vs. pre, $0.05 < P < 0.1$.

stepping in the knee semiflexion 20–30° and kicking. The VI muscle contributed the most (39.6–51.8%) to knee extension (Zhang et al., 2003) and benefited the most from these movements (Cheon et al., 2020; Yang et al., 2019). Furthermore, this study observed a thickening trend in RF and quadriceps (the sum of the thickness of the RF and VI), consistent with previous research indicating that eight weeks of aerobic exercise at 60% to 80% Max VO$_2$ intensity significantly improved the thickness of quadriceps in subjects (Zenith et al., 2014).

By introducing additional weight-loading, the muscles involved in the ankle strategy, such as the tibialis anterior and soleus, are activated to maintain body balance. Previous studies have indicated that wearing weighted vests during exercise increased lower limb muscle mass (Figueroa et al., 2003; Shaw & Snow, 1998). The soleus thickness showed a marginally significant increase between groups, which could be attributed to methodological differences. Previous studies have often quantified overall lower limb muscle mass through techniques such as nuclear magnetic or dual-energy X-ray absorption (Brown, Birge & Kohrt, 1997). In contrast, this study primarily measured the thickness of specific muscles. Additionally, the applied weight in earlier studies was notably higher, reaching up to 10% of the participant's body weight.

### Muscle fiber length and pennation angle

Studies have demonstrated that in addition to inducing significant muscle hypertrophy, aerobic exercise training has the potential to influence muscle structure, especially muscle fiber length and pennation angle. These structural changes can modify contraction properties and enhance muscle function (*Harber et al., 2009*). However, aside from a marginal decrease in the RF pennation angle within the intervention group, fiber lengths and pennation angles of other muscles remained essentially unchanged compared to the baseline, which was in contrast to specific studies in which concentric or eccentric resistance exercise elicited varying degrees of muscle fiber length and pennation angle increments (*Rodriguez-Lopez et al., 2022*; *Seynnes, de Boer & Narici, 2007*). Nevertheless, these results are consistent with those studies in which exercise interventions did not produce significant changes in pennation angle and fiber length.

## Muscle strength

Previous studies have demonstrated that physical activity can increase muscle strength and balance, while significantly reducing the risk of falls and fractures (*Stevens & Olson, 2000*). Moreover, consistent participation in low-impact aerobic walking exercises can significantly improve knee extension torque among active individuals compared with their sedentary counterparts (*Wen et al., 2017*; *Yang et al., 2022*). After the 12-week intervention, no significant change in the knee extension torque was observed in the intervention group. However, the peak torque/weight ratio of knee flexors in the intervention group was significantly improved compared to the control group, consistent with previous studies, indicating that a combination of aerobic and resistance exercises enhances knee muscle strength.

After 12 weeks of intervention, the flexors peak torque/extensors peak torque (H: Q) in the intervention group increased from 0.57 to 0.64. The typical H: Q ratio is approximately 0.6 for individuals without musculoskeletal issues, which tends to decline with age. A lower H: Q ratio may indicate an imbalance in lower limb muscle strength and the occurrence of leg injury (*Evangelidis et al., 2016*). The increased H: Q observed in this study might be associated with increased knee flexion torque. It could be due to the great involvement of the quadriceps in WBDA. To maintain dynamic balance, aid ligaments in preserving joint stability, and ensure even distribution of pressure across the knee joint, the engagement of the quadriceps activates the hamstrings, thus increasing hamstring torque (*Baratta et al., 1988*).

## Functional fitness

Functional fitness is the ability of the human body to perform daily activities independently and safely without fatigue (*Rikli & Jones, 1999*). Enhanced functional fitness is associated with lower abdominal fat (*Chen et al., 2020*) and health-related quality of life in older adults (*Chung et al., 2017*).

One-leg standing is a standard balance indicator, with eyes open or closed in two states. Previous studies have shown that patients unable to sustain one-legged balance for 5 s with their eyes closed are 2.1 times more likely to fall than those who can maintain balance for

more than 5 s (*Vellas et al., 1997*). Additionally, the time spent on SCST was negatively correlated with somatic symptoms and anxiety/insomnia (*Hayashi et al., 2002*). The SCST can enhance the arterial resilience of the supporting leg (*Zhou et al., 2022*), thereby fostering cardiovascular well-being and preventing cardiovascular diseases in older women. These dual benefits represent the significance of SCST as an indicator of balance ability and a predictive measure of fall susceptibility, thus reflecting the overall health status. Significant improvements were observed in SCST following the 12-week intervention, consistent with previous findings (*Filipović et al., 2021*; *Vogler et al., 2009*). Notably, before the intervention, the majority (56%) of participants in the intervention group exhibited SCST durations of less than 5 s, which substantially decreased to 28% after the intervention. Conversely, the control group witnessed an increase in individuals with sub-5-s SCST from 47 to 53%. It highlights that 12-week WBDA can improve static balance in older women and potentially reduce the risk of falls and fractures.

The improvements of 30 s CST were observed in both the intervention and control groups, consistent with previous findings published in the literature (*Marques et al., 2011*). It may be related to sleep quality, mental state and other factors during testing. Nutrition, like dietary protein (*Mareschal et al., 2020*), was also proven to benefit function fitness.

In contrast to previous studies, this study did not yield evidence of WBDA leading to improvement in the 2-min step test and TUG, which could be attributed to prior studies involving frail subjects or older adults suffering from knee osteoarthritis. Notably, the older women in this study who were in better physical condition had greater physical function. Achieving more significant improvement in physical performance could necessitate an extended intervention period (>16 weeks) and more exercise.

## Compliance and attendance rate

Compliance is closely related to the effect of intervention (*Ariza et al., 2019*). This study combined weight-bearing exercise and dance aerobics, yielding an impressive overall compliance rate of 89.19%. Notably, the intervention group exhibited an even higher compliance rate of 90%, surpassing the compliance rates of previous weight-bearing exercise studies (80.4% (*Kim et al., 2018*) and 84% (*Winters-Stone et al., 2014*), respectively). Compared with the attendance rates of 79.7% and 83% in other combined exercise studies using weight-bearing vests, a higher attendance rate of 95.99% was observed in this study. These compliance and attendance outcomes underscore that WBDA was more readily accepted by older women, thereby amplifying the feasibility of the study. The compliance and attendance rates in exercise interventions are pivotal in ensuring the precision and credibility of experimental outcomes. Therefore, future studies should attach importance to their potential implications on results.

## Limitations

This study had certain limitations that should be acknowledged. Firstly, as the research was carried out during the COVID-19 pandemic, recruiting participants was challenging, resulting in a small sample size. In the future, we will expand the sample size to enhance the reliability and accuracy of our results. Secondly, participants may not prefer to stay

with groups during the COVID -19 pandemic. Therefore, they were divided into groups according to their preferences, potentially introducing a significant risk of bias. Finally, the participants were all older women, which restricts the generalizability of the findings to other age groups or sexes. For further studies, broadening the study population to include men aged 60–75 years could provide insights into the impact of WBDA on the lower limb muscles of older adults.

## CONCLUSION

The study concluded that a 12-week weight-bearing dance aerobics could thicken vastus intermedius, increase muscle strength in older women, and improve physical function, thus improving quality of life. Considering that the control group also showed improvements in physical function, the exercise regimen may need to be adjusted for a greater effect.

## ABBREVIATION

| | |
|---|---|
| **PA** | Physical activity |
| **PAR-Q** | Physical Activity Readiness Questionnaire |
| **IPAQ-SF** | International Physical Activity Questionnaire-short form |
| **SB** | Sedentary behavior |
| **LPA** | Light physical activity |
| **MVPA** | Moderate-to- vigorous physical activity |
| **TPA** | Total physical activity |
| **RF** | Rectus femoris |
| **VI** | Vastus intermedius |
| **GM** | Medial gastrocnemius |
| **GL** | Lateral gastrocnemius |
| **H: Q** | Flexors peak torque/extensors peak torque |
| **30 s CST** | 30 s chair stand |
| **CSR** | Chair sit-and-reach |
| **TUG** | Timed up and go |
| **STCT** | Single-legged closed-eyed standing test |
| **BMI** | Body Mass Index |

### Funding

This work was supported by the National Key Research and Development Program of China (2020YFC2003301) and Zhejiang medicine and health science and technology plan (2020KY959). The National Social Science Fund of China (22BTY099) supported the APC. The funders had no role in study design, data collection and analysis, decision to publish, or preparation of the manuscript.

## Grant Disclosures

The following grant information was disclosed by the authors:
National Key Research and Development Program of China: 2020YFC2003301.
Zhejiang medicine and health science and technology plan: 2020KY959.
National Social Science Fund of China: 22BTY099.

## Competing Interests

The authors declare that they have no competing interests.

## Author Contributions

- Xiaoying Peng performed the experiments, analyzed the data, authored or reviewed drafts of the article, and approved the final draft.
- Tang Zhou conceived and designed the experiments, performed the experiments, analyzed the data, authored or reviewed drafts of the article, and approved the final draft.
- Hua Wu conceived and designed the experiments, prepared figures and/or tables, and approved the final draft.
- Yiyan Li conceived and designed the experiments, prepared figures and/or tables, and approved the final draft.
- Jiajia Liu conceived and designed the experiments, prepared figures and/or tables, and approved the final draft.
- Huan Huang performed the experiments, authored or reviewed drafts of the article, and approved the final draft.
- Changshuang He performed the experiments, analyzed the data, authored or reviewed drafts of the article, and approved the final draft.
- Shaoyu Guo performed the experiments, prepared figures and/or tables, and approved the final draft.
- Muyang Huan performed the experiments, analyzed the data, prepared figures and/or tables, and approved the final draft.
- Lei Shi performed the experiments, prepared figures and/or tables, and approved the final draft.
- Peijie Chen conceived and designed the experiments, authored or reviewed drafts of the article, and approved the final draft.
- Minghui Quan conceived and designed the experiments, authored or reviewed drafts of the article, and approved the final draft.

## Human Ethics

The following information was supplied relating to ethical approvals (*i.e.*, approving body and any reference numbers):
The study was approved by the Ethics Committee of Shanghai Institute of Physical Education (Registration No.: 102772020RT096).

## Data Availability

The raw data are available in the Supplemental Files.

## Clinical Trial Registration

The following information was supplied regarding Clinical Trial registration:
Registration No.: ChiCTR2100047187.

## Supplemental Information

Supplemental information for this article can be found online at http://dx.doi.org/10.7717/peerj.17606#supplemental-information.

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
