# Peer review of "Effects of weight-bearing dance aerobics on lower limb muscle morphology, strength and functional fitness in older women"

_PeerJ, doi:10.7717/peerj.17606_

## Round 0.1 · original submission · Major Revisions

Both reviewers raise substantial concerns regarding the methodology (e.g., sample size, training parameters, and description) as well as the validity of the results. Given these important concerns, we recommend substantial revisions to the present manuscript with adequate consideration of reviewer comments.

Please also find attached an annotated PDF from one of the Section Editors.

Reviewer 1 ·

Basic reporting

The English language should be improved to ensure the text can be clearly understood .
Below are some examples of where the text should be improved to make it more readable and understandable:

Line 1: modify the term “basic step”. It is not clear what is meant. Please modify the term throughout the article.
Line 23: add mean age +/- SD
Line 27: don’t use manufacturers here. Use i.e. ultrasound and isokinetic module.
Line 28: “functional fitness” is a very broad term. Please indicate more specifically which test(s) were done.
Line 37-39: please add “compared to non-exercising control group”
Line 69: favorable compared to what?
Line 70-72: compared to what?
Line 78: spell out: “less than three time”. Reword this sentence for better comprehension.
Line 79-80: reword for better readability
Line 85: reword for better readability
Line 91: aerobic or calisthenics? Please stay consistent to avoid confusion
Line 94: It appears that what is being described is not 'calisthenics' but rather an aerobic step exercise. Please change it throughout the manuscript and don’t use the term 'calisthenics'.
Line 97: “increased to increase the load” is the same but written twice.
Line 115: specify what kind of device, i.e. isokinetic device.
Line 113ff: Please expand for better comprehension and correct the units for velocity.
Line 122-123: I assume to the point to which the leg was raised refers to the standing leg. Please add this description.
Line 148, line 155, 162-64: English language should be improved.
Table 2: English language improvements need to be made.
Line 206: change to functional fitness tests.
Line 222-227: English language
Line 230: Use ‘weighted vest‘ instead of ‘weight-bearing vest’.
Line 264 – 267: reword for better comprehension
Line 302: Please clarify what the higher compliance rate compares to?
Line 308-309, 313: Improve English language for better comprehension.

My suggestion is to have a professional fluent in English proofread the manuscript.

Table 2, 3, 4 may be improved by indicating that values are +/- SD.

Experimental design

Except for two minor comments, the methods are mostly well described:
Line 219: Were participants really ‘kneeling’? it does not say so in ‘Methods’. Please clarify or change the wording.
Line 245: Please clarify what is meant by‘centrifugal exercise’.

Validity of the findings

The following improvements can be made:
Line 282ff: Improvements in the SCST were observed in the control group as well. Please elaborate how the improvement in the intervention group is not just an effect due to an adaptation to the exercise.
Line 289ff: Same here: Improvements in the CST were observed in the control group as well. Please elaborate how the improvement in the intervention group is not just an effect due to an adaptation to the exercise.
Line 319-322: Since this is not valid for all muscles trained, please indicate which muscle groups this is valid for and mentioned that also the control group made improvements, thus the exercise regimen may need to be adjusted for a greater effect.

Reviewer 2 ·

Basic reporting

See attachment

Experimental design

See attachment

Validity of the findings

See attachment

Additional comments

See attachment

Annotated reviews are not available for download in order to protect the identity of reviewers who chose to remain anonymous.

---

## Round 0.2 · Minor Revisions

I congratulate the authors on a well improved manuscript. The reviewer have now raised only minor comments, which in my opinion need to be revised before the paper can be accepted. Thank you

Reviewer 1 ·

Basic reporting

The term "aerobics dance" is more appropriate than your previous suggestion. Yet, I suggest to use either "aerobic dance" or "dance aerobics".

The clarity and readability of the article has improved, but there are still several places where further adjustments should be made to improve readability (including but not limited to lines 270, 304-305, 308-310, 363-365). On several occasions, punctuation needs to be adjusted to correct English grammar, and spaces need to be added or removed.

There are two abbreviations which cannot be found elsewhere: AS (line 259), ABAS (line 356).

(Line numbers refer to the "track changes word document".)

In Figure 1, adjust the spacing between letters and parentheses, adjust the wording of the second text box from the top, and adjust the appearance of the "Pre-test" font. Be sure to use upper and lower case letters consistently throughout the figure. I suggest changing the following two text boxes from "Do not change your lifestyle" to "No lifestyle changes" and "Willing group" to "Study participants" or something similar.

Experimental design

no comment

Validity of the findings

no comment

Reviewer 2 ·

Basic reporting

The answers provided by the authors are satisfactory and the changes made have significantly improved the manuscript. I do have some additional (minor) remarks.

First, some additional English language editing is advised.

For example, line 21-24: “The study concluded that compared to non-exercising control group, a 12-week WBAD was show to possibly thicken vastus intermedius, increase muscle strength in elderly women and improve physical function. In addition, this study provides a reference exercise program for older women.” The following elements should be changed: ‘The non-exercising control group’, ‘was shown’, ‘elderly women’ should be at the end of the sentence. Also, why do you mention ‘possibly’ when an increase in thickness was detected (non-significant increases should be referred to as non-significant trends)?

Line 40: should be plural ‘older adults’

Line 85-86: Advise rephrasing the following sentence ‘3) signed informed consent and those who participated voluntarily and promised to complete all experiments’. I assume you allude to participants signing informed consent before participating. I do not feel ‘promising to complete all experiments’ is relevant to the inclusion criteria.

Line 114: ‘list’ should be ‘listed’

Line 270-271: ‘The soleus thickness was a marginal significant between groups…’ Suggest rephrasing to ‘The soleus thickness showed a marginally significant increase…’

Line 356: Wrong abbreviation used: ‘ABAS’ should be WBAD


Additional comments.

Line 46-48: Not all of these studies used benches of the same height, please elaborate if there are specific elements that are expected to have a significantly different impact than WBAD. Otherwise it would suffice to only allude to the fact that the forms of weight bearing exercise incorporated different training components/modalities.

Line 46-49: Do you mean attendance (i.e. percentage of people per training session) or adherence to the training (i.e. percentage of people not dropping out before the end of the training program)?

Line 82-82: I would suggest rephrasing 'failure to meet the requirement…’ to ‘a relatively low amount of physical activity (less than 30 minutes of moderate or higher intensity…’

Line 304-305: ‘The knee extensor muscles thickened, perhaps because completing the exercise required many extensor muscle participants’ It is unclear what you mean by this.

Line 336-337: What was also observed in the control group? This is unclear form the current phrasing.
Statistical analyses: Please also add how you tested for normality of the data.

Experimental design

na

Validity of the findings

na

Additional comments

na

---

## Round 0.3 · Minor Revisions

I think the authors for greatly improving their manuscript. the reviewers still have some minor revisions which need to be addressed before a decision can be made.

Reviewer 1 ·

Basic reporting

The changes made have improved the manuscript.

A few adjustments are recommended for improvement:

The following sentences should be revised to ensure the use of professional English language, which will enhance readability. (E.g. Sentences should not begin with "And".):
"It could be because the WBDA necessitates active involvement of the quadriceps muscles. And to maintain dynamic balance, aid ligaments in preserving joint stability, and ensure even distribution of pressure across the knee joint, the engagement of the quadriceps activates the hamstrings, increasing hamstring torque."

The manuscript contains several instances where spaces are missing before a parenthesis.

Experimental design

no comment

Validity of the findings

no comment

Reviewer 2 ·

Basic reporting

Some additional minor remarks should be addressed to improve the manuscript.

Throughout the manuscript (e.g. lines 6, 20) you refer to ‘a WBDA'. However, the ‘a’ should be removed or you should refer to 'a WBDA program'.

Line 43-44: You allude to the fact that weight-bearing exercise in previous studies incorporated different components. However, the relevance of these differences is not elaborated upon. Therefore, it is hard for the reader to interpret why you mention this. Perhaps my previous concern with this statement was not formulated clearly. Ideally, a short explanation on the fundamental differences in training approaches that may hamper participation or attendance rates should be mentioned or listed. However, for the sake of word limits at least a short caveat should be included mentioning that all these studies have (relatively) high attendance rates despite differences in training approaches.

Line 182: strange formulation ‘achieved all study’, suggest to rephrase.

Experimental design

NA

Validity of the findings

NA

Additional comments

NA

---

## Round 0.4 · accepted · Accept

Thank you for incorporating all suggested comments.